# Early Tree Growth in Reclaimed Mine Soils in Appalachia USA

**Kara Dallaire** [1] **and Jeffrey Skousen** [2,*] 

1   Paragon Soil and Environmental Consulting, Edmonton, AB T5L 2N9, Canada
2   Division of Plant and Soil Sciences, West Virginia University, Morgantown, WV 26506, USA
*   Correspondence: jskousen@wvu.edu; Tel.: +1-304-293-2667

**Abstract:** Surface mining disturbs hundreds of hectares of land every year in many areas of the world, thereby altering valuable, ecologically-diverse forests. Reforestation of these areas after mining helps to restore ecosystem functions and land value. In Appalachia, native topsoil is normally replaced on the surface during reclamation, but waivers allow for brown and gray sandstone materials to be used as topsoil substitutes. Numerous studies report the growth of trees in these substitute mine soil materials, but few studies have compared the height of trees grown in reclaimed mine soils to the heights of trees grown in native soils. This study determined the growth of red oak (*Q. rubra* L.), white oak (*Quercus alba* L.), and tulip poplar (*Liriodendron tulipifera* L.) in two mine soil types which were compared to projected growth in native soils. Heights of tree seedlings in native soils at 11 years were estimated from site indices (SI) from USDA Soil Survey data. At the mine sites, areas with brown and gray mine soils (one site with a mulch treatment) had 12 tree species planted and growth was measured annually for 11 years. Mine soil pH after 11 years was 5.3 for brown mine soils, 6.6 for gray mine soils, 7.0 for mulched mine soils, and 4.1 to 5.2 for native forest soils. After 11 years, tree heights in gray mine soils were significantly lower (0.5 m) than tree heights in brown mine soils (2.8 to 4 m) for all three species. Trees in mulched mine soils were up to 0.7 m taller than trees in un-mulched brown mine soils. After 11 years, red oak height was 6.3 m in native soils and 3 m in brown and mulched mine soils (52% lower); white oak was 7.3 m tall in native soils compared to 3.6 m in brown mine soils (50% lower); and tulip poplar was 11.5 m tall in native soils and 3.5 to 4 m tall in brown and mulched mine soils (70% lower). In gray mine soils, trees were not growing at all. While the trees in brown mine soils are growing, tree growth has not reached projected levels of tree growth in native soils during the first 11 years after planting. The purpose of forestry reclamation is to restore ecosystem diversity and function. This study showed that one measure of ecosystem function, tree growth, was 50% lower on reclaimed mine soils than native forest soils. Maturing mine soils may develop properties over time that are similar to native soils and, with the increased rooting depth, may provide conditions where increased tree growth rates and height may be attained during the next several decades.

**Keywords:** land reclamation; red oak; reforestation; restoration; tulip poplar; white oak

## 1. Introduction

Forested landscapes throughout the world are being severely disturbed by surface mining and other extraction techniques to obtain mineral and energy resources. These disturbances contribute to the need for land reclamation and forest restoration [1]. According to the World Resources Institute [2], 30% of global forest cover has been cleared and an additional 20% has been degraded, with the remaining 50% being fragmented. Many countries have environmental laws that require reclamation and a return of the land to productive land uses after disturbance, but there are many that still do not

have reclamation laws and standards [3,4]. The goal of forest reclamation is to restore the productive capability of the land with an ecosystem that is composed of native species that will function to provide a diversity of economic and ecological values [5]. Studies on forest re-establishment on disturbed areas are widespread such as those found in Australia [6], Canada [7], China [8], Germany [9], New Zealand [10], South Africa [11], and the US [12].

The Central Appalachian ecoregion in the US is an area of about 60,000 km$^2$ and is about 85% covered in eastern deciduous forests. Much of this forested land area is underlain by coal. Surface mining drastically disturbs the landscape and removes these forest ecosystems. Reforestation of mined lands is important because forests provide important benefits to society, including wood and fiber production, watershed and hydrologic values, wildlife habitat, carbon sequestration, and ecosystem stability [12].

Early surface mining operations did little to reclaim the land after mining [13]. Early reclamation efforts simply planted a few trees or filled in excavated pits [14]. The Surface Mining Control and Reclamation Act (SMCRA) was enacted in 1977 by the federal government to address safety and environmental concerns and caused a major change in reclamation practices [4]. Under SMCRA, mined areas were returned to their approximate original contour and topsoil was saved and replaced on the surface. Frequently however, selected overburden materials replaced topsoil when the substituted material was found to be equal to or better for sustaining vegetation as the topsoil or when insufficient amounts of topsoil were available. Grasses and legumes were often chosen to be planted because they controlled erosion and provided quick economic returns to the landowner and they grew adequately in the substitute topsoil [15]. Reforestation had previously been an important reclamation strategy for mined lands, but SMCRA reclamation requirements unintentionally shifted the emphasis for revegetation from trees to herbaceous covers. The herbaceous cover crops covered the ground successfully and caused severe competition to other species, particularly trees and invasive plants, that were recruited to the site [16]. Between 1977 and 2000, few mined areas in the US were reclaimed with trees [13] and trees were slow to naturally re-colonize reclaimed pastures, resulting in few areas being restored to Appalachian forest ecosystems after the enactment of SMCRA [17].

In response to this loss of forests after mining and reclamation, the Appalachian Regional Reforestation Initiative (ARRI) was created in 2003 to encourage restoration of high quality forests on reclaimed mines [18]. Scientists with ARRI developed the Forestry Reclamation Approach (FRA) to encourage forest re-establishment after mining under SMCRA [19]. The FRA is composed of five steps:

1. Create a suitable rooting medium for good tree growth that is no less than 1.2 m deep and comprise topsoil, weathered sandstone and/or the best materials.
2. Loosely grade the topsoil or topsoil substitute established in step one to create a non-compacted growth medium.
3. Use ground covers that are compatible with growing trees.
4. Plant early successional trees for wildlife and soil stability, and commercially valuable trees.
5. Use proper tree planting techniques.

Due to steep terrain, mining companies have difficulty collecting and saving the relatively thin layer of native topsoil in the Appalachian region. Weathered brown sandstones and unweathered gray sandstones that are exposed during mining can be used as substitute topsoil materials [20]. Brown sandstone materials are found closer to the surface and typically have a pH ranging from 4.0 to 5.5 due to weathering [21]. Brown sandstone has low electrical conductivity (EC) and high percentages of fines (<2 mm); both of which correlate with increased tree growth [22,23]. In this paper, brown sandstone overburden placed at the surface for revegetation is referred to as "brown mine soil." Gray sandstone overburden is found at lower depths and underneath brown sandstone overburden in the geologic column and generally has a high alkaline earth composition. Without weathering and release of the leachable bases, the pH of gray sandstone material ranges from 7.5 to 8.0 [21,24]. "Gray mine soil" in this paper refers to the unweathered gray sandstone material placed at the surface as topsoil substitutes for revegetation. Brown mine soil is the preferred topsoil substitute in reforestation

projects when insufficient quantities of native soil are salvaged [20] and numerous studies have shown that brown mine soil promotes superior tree growth over gray mine soil [24–29].

Tree growth can be improved with amendments on reclaimed sites [25,28,30]. In a study by Angel et al. [25], the addition of bark mulch improved the growth of two tree species on uncompacted plots on a reclaimed coal mine in eastern Kentucky. Omari et al. [31] found improved tree growth with amendments in reclamation soils of Canadian oil sands rehabilitation. Wilson-Kokes et al. [28] found that adding bark mulch to both brown and gray sandstone increased the growth of hardwood species on a reclaimed coal mine in West Virginia.

Site index (SI) is defined as the total height of a specific species of tree at 50 years of age. It is a standard measure used throughout the world as an estimate of the relative productivity of forest land, which is calculated from height measurements of free-growing, uninjured, dominant, and co-dominant trees in a fully stocked stand [32,33]. Site index curves are specific to species, geographic region, and the soil and/or topography [34]. Soil properties related to SI are surface soil depth, subsoil texture, aspect, and slope position and steepness [35] with higher SI found on sites with favorable soil properties and site conditions for tree growth. It is hypothesized that mine soils have lower SI due to their lower soil quality for tree growth, particularly when gray sandstone is used.

The purpose of this study was to determine whether tree growth on mine soils is equal to growth in native forest soils. This study compared the height growth of red oak, white oak, and tulip poplar grown in two mine soil types on two reclaimed mines in West Virginia to estimated heights of the same species in native forest soils. The heights of tree seedlings in native soils 11 years after planting were estimated from heights based on the pre-mine SI for native soils at each mine site from the USDA-NRCS Web Soil Survey.

## 2. Materials and Methods

Tree seedling heights in mine soils were determined at two sites. The first site was Birch River (BR), owned by Arch Coal and located in Webster County, WV (38°26′27.56″ N 80°36′27.55″ W). Three 4 ha plots were established in November 2006; the first was composed of weathered brown mine soils, a second of unweathered gray mine soils, and a third with a brown/gray mine soil mix overlain by 15 cm of bark mulch. The bark mulch contained ground up limestone, which was placed as aggregate at the sawmill landing and was incorporated into the bark waste before hauling to the mine site. In March 2007, 12 tree species were planted at 2.5 m centers for a stocking rate of about 1450 trees per ha (full site details and planting methods are available in [28]). At BR, tree growth was measured annually between 2007 and 2015. To determine tree height, 11 2.7 m wide transects, each spanning the experimental plots, were used to determine tree height [28].

The second site was Catenary (CY), owned by Patriot Coal and located in Kanawha County, WV (38°02′45.11″ N 81°30′32.38″ W). Two 2.5 ha plots were constructed in January 2005 by placing weathered brown sandstone or unweathered gray sandstone on the surface as mine soil. In March 2005, 11 species of hardwood tree seedlings were planted by a professional planting crew on 2.3 m centers (full site details are available in [29]). At CY, trees were sampled annually between 2005 and 2015 with two 2.7 m wide by 195 m long transects. For both mine sites, any tree within transects was identified by species and height to the highest live growth was measured and recorded. For this study, data for only red oak, white oak, and tulip poplar were used. The number of trees sampled for each species to determine the average height varied from 37 to 74 for red oak, 37 to 69 for white oak, and 24 to 27 for tulip poplar at the BR and CY sites, respectively [28,29].

Tree seedling heights in native soils after 11 years were estimated by pre-mine SI at each of the mine sites from the USDA-NRCS Web Soil Survey [36]. The pre-mining soils at the BR site were from the Gilpin-Buchanan complex, which are classified as fine-loamy, mixed, mesic Typic Hapludults. Pre-mining soils at the CY site were from the Clymer-Dekalb complex, which were also classified as fine loamy, mixed, mesic Typic Hapludults. Pre-mine SI for each mine site was determined by calculating a weighted average of soil types occurring on the site and determining corresponding SI

listed for those soils in the Web Soil Survey. Once an average SI was calculated at each site for northern red oak, white oak, and tulip poplar, this number was used to estimate the height of trees from 1 to 11 years of age. To calculate the estimated height for each year, the formulation equation from Carmean et al. [32] was used:

$$H = b_1 S^{b_2 (1 - e^{b_3 A})} b_4 S^{b_5} \tag{1}$$

H = Height
(1–4) = Regression Parameters
S = Site Index
A = Age

The pre-mine SI for native soils at BR at 50 years was 24 m for both red oak and white oak, and 28 m for tulip poplar (Table 1; [36]). At CY, the SI for native soils at 50 years was 24 m for red oak, while white oak and tulip poplar had higher SI at 26 m and 29 m, respectively (Table 1; [36]). Table 2 lists the study sites and treatments along with their abbreviations.

**Table 1.** Pre-mine site index (SI) height in meters at 50 years for red oak, white oak, and tulip poplar at the Birch River (BR) and Catenary (CY) mines in West Virginia.

| Species | BR SI | CY SI |
|---|---|---|
| | — m — | |
| Red oak | 24 | 24 |
| White oak | 24 | 26 |
| Tulip Poplar | 28 | 28 |

**Table 2.** Pre-mining native soil abbreviations and brown and gray mine soil abbreviations.

| Site-Treatment | Abbreviation |
|---|---|
| Pre-mining Birch River | P-BR |
| Pre-mining Catenary | P-CY |
| Birch River mulch | BR-M |
| Birch River brown sandstone | BR-B |
| Birch River gray sandstone | BR-G |
| Catenary brown sandstone | CY-B |
| Catenary gray sandstone | CY-G |

Soil data at the mine sites were taken from Wilson-Kokes et al. (28 m [29]) and reported here to describe mine soil properties. Soil data were analyzed by one-way ANOVA by site for the 2012 sampling year. Soil properties for the native soil sites were taken from the USDA Web Soil Survey [36].

Tree height data from the BR and CY mine sites were analyzed using regression with site, year, and species to model annual tree growth rates over time. The slopes of these regression equations on the mine sites for each species were analyzed by ANOVA to determine differences among mine soil types. When significant differences were found with ANOVA ($p < 0.05$), differences among means were computed with Tukey's Honestly Significant Difference test at a level of $p \leq 0.05$. All statistical analyses were performed using the statistical program R [37].

## 3. Results and Discussion

Table 3 lists the pH of native forest soils at Pre-mining Birch River (P-BR) and Pre-mining Catenary (P-CY), which varied from 4.1 to 5.2 [36], a common range for soils in Appalachian forests [21,38–40]. For mine soils, the pH was higher, ranging from 5.2–5.4 for the brown mine soils at Birch River-brown sandstone (BR-B) and Catenary-brown sandstone (CY-B), and 6.5 to 6.8 for gray mine soils at Birch River-gray sandstone (BR-G) and Catenary-gray sandstone (CY-G). Due to limestone being mixed with bark mulch at the sawmill, the pH of mulched mine soils (Birch River-mulch, BR-M) was 7.0.

Some studies show gray sandstone mine soils to have pH from 7.5 to 8.0 [21,24] and the gray mine soils at BR and CY were lower than 7.5 probably because they had several years of weathering and organic matter addition, both of which tend to lower the pH of mine soils. Values for EC were low and fines ranged from 42% to 60% in the mine soils. The textures of the mine soils ranged from sandy clay loams and sandy loams to loamy sands (data not shown). Measurements of organic matter content of these mine soils were not performed, however studies on similar mine soils after 10 years in Kentucky showed less than 1% total C [41].

**Table 3.** Soil properties (pH, electrical conductivity (EC), percent fines) for pre-mine soils at each site (P-BR and P-CY), Birch River (BR-M, BR-B and BR-G) and Catenary (CY-B and CY-G) mine soils (means ± one standard error).

| Site-Treatment | Properties | | |
| --- | --- | --- | --- |
| | **pH** | **EC** | **Fines** |
| | | $(dS\ m^{-1})$ | (%) |
| P-BR | 4.6-4.7 | 0.0 | NA [1] |
| P-CY | 4.7-4.9 | 0.0 | NA |
| BR-M | 7.0 ± 0.3 a [2] | 0.04 ± 0.03 a | 43 ± 6 b |
| BR-B | 5.2 ± 0.6 c | 0.01 ± 0.03 a | 45 ± 5 b |
| BR-G | 6.5 ± 0.4 b | 0.01 ± 0.03 a | 49 ± 6 b |
| CY-B | 5.4 ± 0.5 c | 0.01 ± 0.02 a | 60 ± 5 a |
| CY-G | 6.8 ± 0.3 ab | 0.01 ± 0.03 a | 42 ± 7 b |

[1] Not available. [2] Values for each property (columns) with the same letter are not significantly different at $p \leq 0.05$.

Extractable nutrient information was only available for the mine soils (Table 4). Calcium, Mg, and P concentrations between gray and brown sandstone mine soils and between the sites were similar. Mulched mine soils (BR-M) had much higher Ca, Mg, and K concentrations (from 2 to 20 times higher) than untreated mine soils. Mine soils with lower pH (BR-B and C-B) had higher Al concentrations than other mine soils (415 versus 130 mg kg$^{-1}$), while the C-G mine soil had almost 5 times higher P concentration than the other mine soils (180 versus 35 mg kg$^{-1}$). The high P concentration in C-G mine soils was documented by other studies [24]. Skousen and Emerson [42] found that this P extracted by Mehlich 1 in these mine soils was not released by weaker extractants that more-closely estimated plant-available P and, therefore, the P in these mine soils was not readily available for plant uptake.

**Table 4.** Extractable nutrients (means ± one standard error). from mine soils at Birch River (BR) and Catenary (CY).

| Site-Treatment | Element | | | | | |
| --- | --- | --- | --- | --- | --- | --- |
| | **P** | **K** | **Ca** | **Mg** | **Al** | **Fe** |
| | mg/kg | ————$cmol_c$/kg———— | | | ——mg/kg—— | |
| BR-M | 23 ± 15 b [1] | 12 ± 6 a | 197 ± 22 a | 12 ± 4 a | 115 ± 14 b | 51 ± 15 c |
| BR-B | 38 ± 20 b | 0.6 ± 0.4 b | 6 ± 4 b | 4 ± 1 b | 402 ± 55 a | 219 ± 20 a |
| BR-G | 46 ± 22 b | 0.3 ± 0.3 b | 6 ± 2 b | 4 ± 2 b | 113 ± 18 b | 126 ± 28 b |
| CY-B | 41 ± 18 b | 0.6 ± 0.3 b | 7 ± 1 b | 5 ± 1 b | 431 ± 62 a | 154 ± 21 b |
| CY-G | 180 ± 38 a | 0.4 ± 0.2 b | 10 ± 2 b | 5 ± 2 b | 150 ± 23 b | 149 ± 14 b |

[1] Values for each element (columns) with the same letter are not significantly different at $p \leq 0.05$.

Red oak was calculated to attain a height of 6.3 m after 11 years in pre-mine native soils at the BR and CY sites (Figure 1). The CY forestry study was established in 2005 and therefore had 11 years of data, while the BR study was established in 2007 and had only nine years of data. Therefore, height trendlines extend to 11 years for CY and only nine years for BR. Red oak trees on brown mine soils (BR-B and CY-B) had lower growth rates (reaching about 3 m after nine and 11 years) than red oak height in native soils (6.3 m) or about 50% less height. Red oak trees in mulched mine soil (BR-M)

showed very similar growth to the brown mine soil (BR-B) with about 3 m of growth after nine years. Red oaks in gray mine soils at both sites (BR-G and CY-G) were clearly growing at slower rates than the other sites (0.5 to 0.8 m in height).

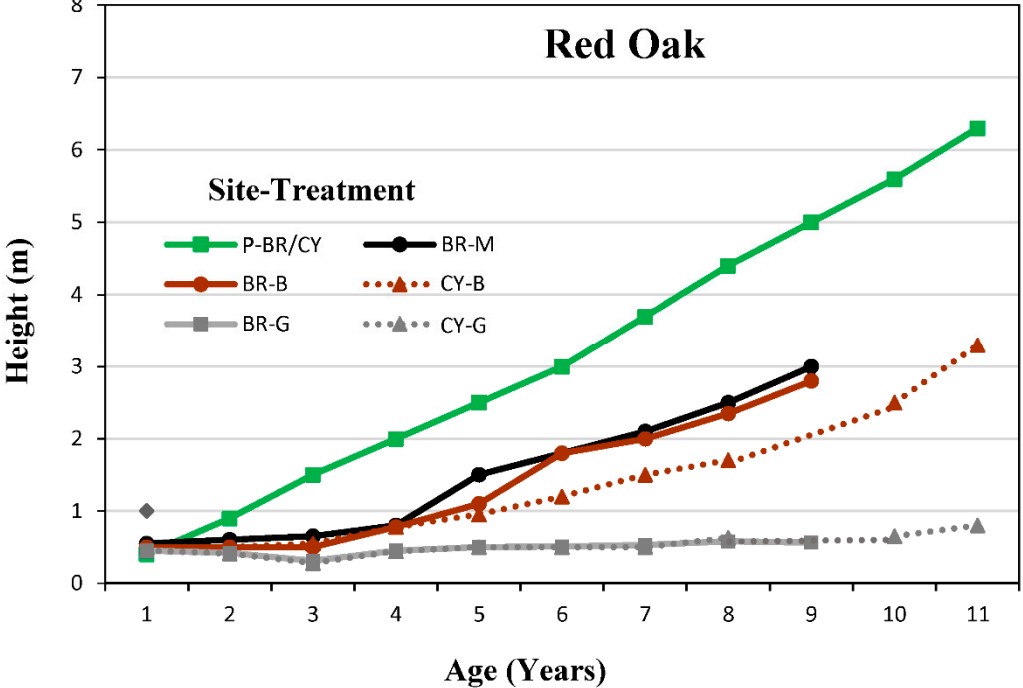

**Figure 1.** Height of red oak during the first 11 years of growing in pre-mine native soils (P-BR/CY) and in brown (BR-B and CY-B) and gray mine soils (BR-G and CY-G) at two reclaimed mine sites.

Linear regression equations were developed for red oak height growth over time from the data plotted in Figure 1 (Table 5). Significantly different rates of tree growth were found among the mine soil types. Red oak growth rates on pre-mine native soils were 0.6 m per year, which was about 50% higher than those in the brown mine soils at around 0.3 m per year. Gray mine soils showed almost no growth at all.

**Table 5.** Regression equation values for red oak growing in pre-mine native soils (P-BR/CY) and in brown (BR-B and CY-B) and gray mine soils (BR-G and CY-G) at two reclaimed mine sites.

| Site-Treatment | Slope | Intercept [1] |
|---|---|---|
| | - m per year - | - m - |
| P-BR/CY | 0.593 | 0.28 |
| BR-M | 0.323 a [2] | 0.12 |
| BR-B | 0.313 a | 0.19 |
| CY-B | 0.268 a | 0.18 |
| BR-G | 0.024 b | 0.36 |
| CY-G | 0.038 b | 0.30 |

[1] Intercepts were not analyzed by ANOVA. [2] Slope values with the same letter are not significantly different at $p \leq 0.05$.

White oak height was 5.5 to 7.3 m after 11 years on the native soils at both sites with SI of 24 m and 26 m (Figure 2). White oak attained slightly higher height values (7.3 m) than red oak (6.5 m) on pre-mine native soils. In brown mine soils, white oak seedlings attained heights of 2.3 m at BR-B and CY-B at nine and 10 years, respectively. An increased growth rate appeared for white oak between years nine and 11 at CY-B, and red oak showed this similar growth trend at CY-B (Figure 1). White oak growth

in mulched mine soils (BR-M) had better height growth than BR-B (25% greater), which could be due to its response to greater amounts of nutrients and water [43]. The gray mine soils at both sites (BR-G and CY-G) had very poor growth with heights of only 0.4 to 0.8 m after nine and 11 years, respectively.

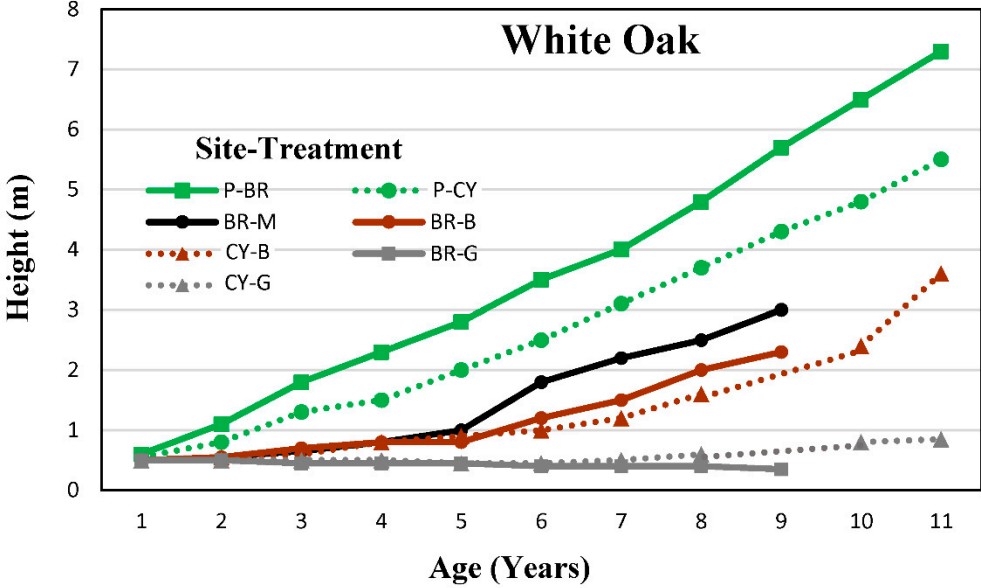

**Figure 2.** Height of white oak during the first 11 years of growing in pre-mine native soils (P-BR and P-CY) and in brown (BR-B and CY-B) and gray mine soils (BR-G and CY-G) at two reclaimed mine sites.

White oak growth rates were 0.66 m per year in BR pre-mine native soils compared to 0.25 m per year in BR brown mine soils (about 65% less) and 0.025 m per year in gray mine soils (96% less) (Table 6). Growth rates were slightly greater for white oak on pre-mine native soils than red oak (0.66 versus 0.59 m per year; Tables 5 and 6), and the brown and gray mine soils showed similar height growth rates as red oak (0.301 ave white oak versus 0.276 ave red oak m per year for brown, respectively).

**Table 6.** Regression equation values for white oak growing in pre-mine native soils (P-BR and P-CY) and in brown (BR-B and CY-B) and gray mine soils (BR-G and CY-G) at two reclaimed mine sites.

| Site-Treatment | Slope | Intercept [1] |
|---|---|---|
|  | - m per year - | - m - |
| P-BR | 0.663 | 0.31 |
| P-CY | 0.502 | 0.28 |
| BR-M | 0.333 a [2] | 0.22 |
| BR-B | 0.226 a | 0.02 |
| CY-B | 0.269 a | 0.22 |
| BR-G | 0.017 b | 0.52 |
| CY-G | 0.033 b | 0.37 |

[1] Intercepts were not analyzed by ANOVA. [2] Slope values with the same letter are not significantly different at $p \leq 0.05$.

The height growth of tulip poplar was greater than the two species of oak (Figure 3). Tulip poplar height was 11 m to 12 m on pre-mine native soils after 11 years compared to 6 m to 7 m for the oaks (Figures 1 and 2). Tulip poplar trees on mine soils achieved heights of almost 4 m (65% less) after nine years in BR-M and BR-B, and 3.3 m (70% less) after 11 years in CY-B mine soils. Tulip poplar trees grew poorly on both gray mine soils and, in fact, appeared to be declining in height. Similar to this study, tulip poplar was found to grow 36% greater than that of red oak under combined high levels of light, moisture, and nutrients [44]. Tulip poplar tends to be more opportunistic and adapts to productive environments, while red oak tends to be adapted to moderately unproductive environments.

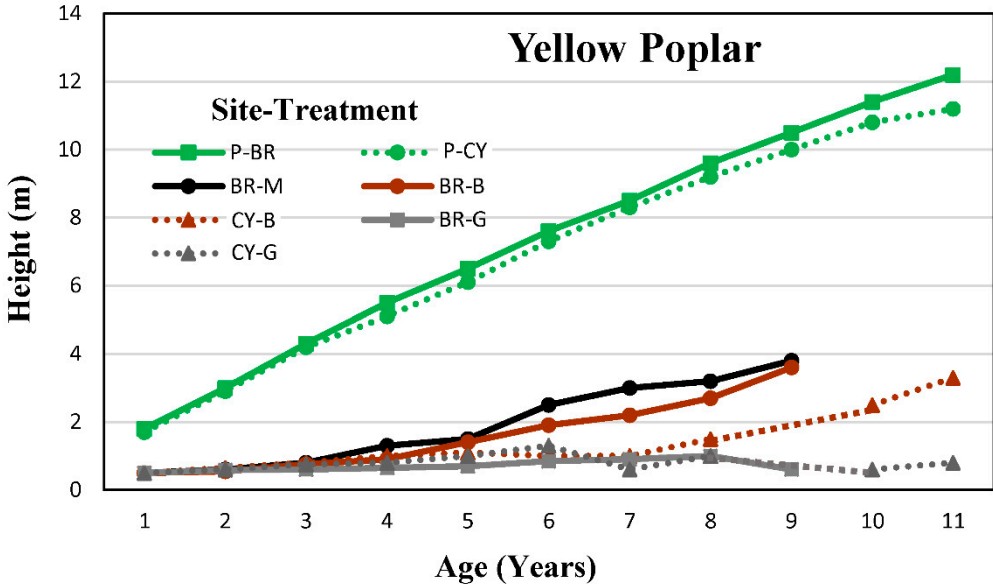

**Figure 3.** The height of tulip poplar growing in pre-mine native soils (P-BR and P-CY) and in brown (BR-B and CY-B) and gray mine soils (BR-G and CY-G) at two reclaimed mine sites.

Annual growth rates for tulip poplar averaged about 1 m per year in pre-mine native soils (Table 7). For brown and mulched mine soils (BR-M, BR-B and CY-B), tulip poplar grew 0.24 to 0.44 m per year (65% to 75% lower). And the gray mine soils (BR-G and CY-G) showed no growth and had the lowest growth rates (0.02 m to 0.04 m per year) of any mine soil type.

**Table 7.** Regression equation values for tulip poplar growing in pre-mine native soils (P-BR and P-CY) and in brown (BR-B and CY-B) and gray mine soils (BR-G and CY-G) at two reclaimed mine sites.

| Site-Treatment | Slope | Intercept [1] |
|---|---|---|
| | - m per year - | - m - |
| P-BR | 1.040 | 1.11 |
| P-CY | 0.972 | 1.15 |
| BR-M | 0.443 a [2] | 0.31 |
| BR-B | 0.379 a | 0.28 |
| CY-B | 0.239 b | 0.03 |
| BR-G | 0.042 c | 0.50 |
| CY-G | 0.016 c | 0.70 |

[1] Intercepts were not analyzed by ANOVA. [2] Slope values with the same letter are not significantly different at $p \leq 0.05$.

The purpose of this study was to answer the question, "Do trees planted in reclaimed mine soils grow at rates equal to or better than those growing in undisturbed native forest soils?" During the first nine to 11 years after planting in mine soils, the results showed that these species were not growing at equal or better rates. In brown mine soils, both red and white oak grew at about 50% of the rate as they could grow in native soils using pre-mining SI values. For tulip poplar, rates of growth in brown mine soils were only about 30% to 40% of the growth rate in pre-mining native soils. In gray mine soils at both BR and CY, the trees were not growing at all. The use of bark mulch on these mine soils improved growth rates slightly over that of brown mine soils and significantly over gray mine soils. Cotton et al. [45] compared the growth of white oak and tulip poplar in reclaimed mine soils in Kentucky to the growth of trees in reference forests after five years of growth. They showed hat tree growth was less than 50% on mine sites compared to native soils, which is confirmed by the results of this study.

To make a more effective comparison of early tree growth between native forest soils and mine soils, tree seedlings of the same stock should have been planted at the same time in the brown and gray mine soils as well as in adjacent native forest soils cleared from forest canopy trees. However, this was not done in this study (nor in any study of which we are aware in the Appalachian Region) and we used projections of tree growth from the SI of pre-mining native soils at each site to compare to tree growth in mine soils. This estimate using SI provides a prediction of possible or potential tree height at 11 years.

However, estimating tree height from the SI of pre-mining native soils has drawbacks. Estimated tree heights of three species were based on the SI height at age 50 (not at age 11), and annual growth rates were calculated by the equations in Carmean et al. [32]. Growth rates of trees are not constant, especially during the early years, and height growth is often characterized by four growth phases over the life-cycle of trees [45–47], with the early growth phase often being more rapid than later years. Carmean et al. [32] noted that predicting heights of younger trees tends to be less accurate than when estimating heights of 30- to 50-year-old trees.

As another comparison using field monitoring data, seedling heights for red oak and tulip poplar at 11 years were available from a clear-cutting study at the Fernow Experimental Forest in West Virginia [48,49]. The SI for red oak at the Fernow was 23 m (similar to the SI of 24 m in pre-mining native soils at BR and CY), and seedling heights measured after 11 years in the Fernow were 4.7 m. Therefore, red oak seedlings growing in brown mine soils were about 30% lower than the heights at the Fernow (3.3 m versus 4.7 m). The SI for tulip poplar at the Fernow Forest was 42 m (SI was 28 m in pre-mining native soils at BR and CY) and seedling heights measured in the Fernow at age 11 were 9 m. Tulip poplar trees growing in brown mine soils attained a height of 4 m after 11 years and therefore the height was about 55% lower in mine soils compared to tulip poplar trees in the Fernow Forest for the same growth period (4 m versus 9 m). These height values from the Fernow at 11 years further demonstrate the lower rates of growth of these trees in mine soils versus native forest soils.

Researchers in central Ohio planted red oak seedlings into a pasture field and the trees attained heights after eight years of 5 m to 6.9 m [50]. These measured heights in Ohio were higher than the heights estimated by the SI in native soils at our two sites (Figure 1 shows red oak heights to be 4.5 m at eight years with the Web Soil Survey SI values). In northern West Virginia, Brooks [51] found planted tulip poplar seedlings to average 5.5 m in height seven years after planting in native field soils. Based on projected growth in that paper, these trees would reach greater than 12 m at 11 years, and the height was similar to the projected height estimates for our pre-mining soils at both sites.

Will trees in mine soils *always* grow slower than trees growing in native soils? Based on observations of older mined and reforested sites after two or three decades, mine soils developed into suitable plant growing materials and trees grew as well or better than those in surrounding undisturbed forests [14,52]. For example, black walnut (*Juglans nigra*) at 47 years of age in mine soils attained heights that were greater than the highest reported SI for black walnut at age 50 in Illinois forest soils [53].

The natural soils in undisturbed forests developed slowly from parent material in response to the soil-forming factors [54]. Due to steep slopes in Appalachia, these soils have eroded and are relatively shallow on ridgetops and side slopes and consequently have restricted rooting depths for trees. Surface mining and reclamation removed the shallow native soils and replaced them with blasted rock materials overlain by a topsoil material (whether replaced native soil or substitute brown or gray mine soils). In doing so, the restricted rooting depth of pre-mining native soils has been eliminated and the new mine soils have greater depth and volume for root development. One of the important determining soil properties for SI is surface soil depth [35]. Therefore, by altering one of the most important SI properties (soil depth) by surface mining and reclamation, and as mine soil properties return over time to their original condition through weathering and development, trees may grow in mine soils at greater rates and may achieve greater heights than those predicted by SI of pre-mining native soils. As fresh unweathered gray materials age, they release nutrients that become available for plant uptake. Access to those nutrients increases as roots explore greater volumes of soil with

maturation, and these improved soil conditions may continue for the next several timber rotations [55]. Trees on maturing mine soils with unrestricted rooting depth may have the potential of increased growth during the next century or more.

## 4. Conclusions

This study was conducted to answer the question, "Do native trees planted in mine soils grow at rates equal to or better than trees planted in native forest soils?" Based on the results of this study, the answer is no. Tree heights for three tree species in mine soils after nine and 11 years were lower than heights estimated from pre-mining native forest soils. Growth rates for red and white oak on brown mine soils had growth rates that were about 50% less than that of trees growing on native soils. Tulip poplar growth rates on brown mine soils were about 70% less than the rates of growth on native soils. In gray mine soils, trees were not growing. Mulching improved the growth rates of trees in mine soils, but the growth rate was only slightly better than those growing in brown mine soils. With time, new mine soils will evolve as they weather to finer soil-sized particles, acquire and sequester organic matter, develop greater microbial numbers and diversity, and cycle nutrients. Through this process, mine soils may develop properties over time that are similar to native soils and, with the increased rooting depth and nutrients, may provide conditions where increased tree growth rates above those predicted by the SI may be attained. Further studies of tree height 20 to 30 years after planting on these mine soils will help determine the trajectory of growth and will help evaluate improvements in mine soil health over time. The purpose of forestry reclamation is to restore ecosystem diversity, function, and sustainability. In the short term, these purposes were not met, and this study showed that one measure of ecosystem function, tree growth, was 50% lower in reclaimed mine soils than that of native forest soils. However, the placement of suitable soil materials at the surface and the planting of native trees may have begun the process of forest community development, and these newly-formed forest communities may evolve and progress into a functioning and sustainable forest with the passage of time.

**Author Contributions:** J.S. conceived and designed the experiments; K.D. performed the experiments and analyzed the data. Both authors wrote the paper.

**Funding:** This research was funded by funds provided by the Hatch Act at West Virginia University, Arch Coal-Birch River, and Catenary Coal.

**Acknowledgments:** The authors would like to thank past graduate students who monitored and provided some of the data for this study including Paul Emerson, Curtis DeLong, Calene Thomas, and Lindsay Wilson-Kokes. Appreciation is also expressed to Keith O'Dell and John McHale for providing support and access to the sites. Help was also provided by Jamie Schuler, Louis McDonald, and Paul Ziemkiewicz of West Virginia University. Scientific Article No.3362 of the West Virginia Agricultural and Forestry Experiment Station, Morgantown.

**Conflicts of Interest:** The authors declare no conflict of interest. The funders had no role in the design of the study; in collections, analyses, or interpretation of data; in the writing of the manuscript; or in the decision to publish the results.

## Abbreviations

| | |
|---|---|
| ARRI | Appalachian Regional Reforestation Initiative |
| FRA | Forestry Reclamation Approach |
| SMCRA | Surface Mining Control and Reclamation Act |

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
