# Peer review of "Early Tree Growth in Reclaimed Mine Soils in Appalachia USA"

_forests, doi:10.3390/f10070549_

Round 1

Reviewer 1 Report

As a restoration ecologist working on the science and practice behind returning biodiverse plant communities to post-mining landscapes following mining disturbance, I found this study interesting but rather pedestrian and very simplistic. Essentially, this study just compares estimated tree height indices for three species after time in post-mining areas - with a little side information on soil chemistry - to infer revegetation trajectory on different mine substrates.

It is in almost every sense parochial; the title, for example, makes no sense to someone not familiar with local geomorphology: what are "brown and gray mine soils"? I see that these terms have been published undefined by the authors previously, e.g., Wilson-Kokes et al 2013, Emmerson et al 2009, and even Wilson-Kokes and Skousen 2014 in STOTEN. However, even understanding that, from the methods, it refers to different coloured Sandstone, it is rather meaningless. Many soils around the world are brown or gray. Surely the authors can present a proper geological classification and terminology.

"Soil" is not an appropriate term for heavily disturbed post-mining materials, either, in this context. Post-mining wastes are heavily altered and not representative of soils.

"Site indices" are a local term that have little meaning to an international audience, and are not comparable to studies published in other regions. It would be useful if the authors could either clearly define the value of 'site index' in the context of the international literature, or select another metric.

Much of the background and literature cited focusses exclusively on the Appalachian region, and does not contextualise the results in a global or even national sense. The terminology used throughout is local (understanding it is used within SMCRA and FRA) and not up to step with internationally-accepted norms, see for example the International Standards for the Practice of Ecological Restoration (SER 2016) which clearly articulate the difference between different post-mining activities, and recent commentaries such as "Cross AT, Young R, Nevill P, McDonald T, Prach K, Aronson J, Wardell-Johnson G, Dixon KD. 2018. Appropriate aspirations for effective post-mining restoration and rehabilitation: a response to Kaźmierczak et al. Environmental Earth Sciences 77, 256" which illustrate that "reclamation" is a term no longer widely used by the international scientific community. This study examines rehabilitation, and should utilise the appropriate terminology to reflect this if the study is to be internationally relevant.

The statistical approach is not sufficient, or at least not sufficiently described in the methods. How were linear regression analyses undertaken? With what software? Utilising what factors or variables, and under what conditions? Were underpinning assumptions of normality, heteroscedasticity, and linearity tested and confirmed? Which hypotheses were tested, and how were analyses set up in order to achieve this? Why were ANOVA undertaken on regression coefficients, and not on raw data? What kind of ANOVA? Were post-hoc tests utilised, as it appears from later tables? Were data transformed in any way prior to analysis? If so, why and which transformation? It is inapropriate to simply say "P <0.05" to indicate statistical significance - where are the F values and d.f. for ANOVA, and other regression details from analyses? At the very least, parochial issues aside, the statistical deficiencies must be addressed prior to publication.

Figures are created in excel, and are not of sufficient quality for publication.

Figure and table captions do not contain sufficient information (e.g., details of statistical tests used to infer significance between and among treatments).

Many of the references are very old, and in some cases out of date with the current literature.

Both author affiliations are given as 1, despite each having a different affiliation.

I have numerous other comments on specific elements of the manuscript and its content (e.g., grammar and spelling issues throughout), but these are relatively minor in the context of the more significant issues relating to statistical treatment and interpretation of the data. While I am not against the publication of parochial studies, and this study could be published with work, it is not good enough in its current form. Of particular note for me was the authors own comment on line 272: "A better experimental design would have benefited this study".

Author Response

Responses to Reviewer Comments

Forests-520465

Early Tree Growth in Reclaimed Mine Soils in Appalachia USA

Dallaire and Skousen

Dear Editor:

Thanks again for getting critical reviews of this paper. We will answer the questions posed by the reviewers and show the changes we made in the manuscript.

Reviewer 1

1. The title, for example, makes no sense to someone not familiar with local geomorphology: what are "brown and gray mine soils"? I see that these terms have been published undefined by the authors previously, e.g., Wilson-Kokes et al 2013, Emerson et al 2009, and even Wilson-Kokes and Skousen 2014 in STOTEN. However, even understanding that, from the methods, it refers to different coloured Sandstone, it is rather meaningless. Many soils around the world are brown or gray. Surely the authors can present a proper geological classification and terminology.

Response: Agree.

The title of the paper has been changed:

Early Tree Growth in Reclaimed Mine Soils in Appalachia USA

We have made a better definition and description of brown and gray mine soils as topsoil substitutes. The distinction between these two types of mine soils is important for understanding the results of this study. The study came about because regulators asked me how tree growth in mine soils compared with growth in native forest soils. That question has not been answered well, and this study attempts to address it.

The wording in the manuscript has been changed to:

“…Weathered brown sandstones and unweathered, gray sandstones that are exposed during mining can be used as substitute topsoil materials (Skousen et al. 2011). Brown sandstone materials are found closer to the surface and typically have a pH ranging from 4.0 to 5.5 due to weathering (Haering et al. 2004). Brown sandstone has low electrical conductivity (EC) and high percentages of fines (<2 mm); both of which correlate with increased tree growth (Daniels and Amos 1985; Rodrigue and Burger 2004). In this paper, brown sandstone overburden placed at the surface for revegetation is referred to as ‘brown mine soil.’ Gray sandstone overburden is found at lower depths and underneath brown sandstone overburden in the geologic column and generally has a high alkaline earth composition. Without weathering and release of the leachable bases, pH of gray sandstone material ranges from 7.5 to 8.0 (Haering et al. 2004; Emerson et al. 2009). ‘Gray mine soil’ in this paper refers to the unweathered gray sandstone material placed at the surface as topsoil substitutes for revegetation. Brown mine soil is the preferred topsoil substitute in reforestation projects when insufficient quantities of native soil are salvaged (Skousen et al. 2011) and numerous studies have shown that brown mine soil promotes superior tree growth over gray mine soil (Angel et al. 2006; Emerson et al. 2009; Sena et al. 2014; Showalter et al. 2010; Wilson-Kokes, et al. 20013a, 2013b).”

2. "Soil" is not an appropriate term for heavily disturbed post-mining materials, either, in this context. Post-mining wastes are heavily altered and not representative of soils.

Response: This point has been discussed in the soil science and reclamation literature for over 50 years. The consensus in the US is that material placed or ending up on the surface after disturbance is “soil” since it is the material in which plants will colonize and grow. Adjectives can be applied to the term soil such as “dredged soils,” “mine soils,” “urban soils,” “sub soils,” “flooded or eroded soils,” “deposited soils,” anthropomorphic soils,” etc. Therefore, the term soil is appropriate for the final layer of earth-like material placed on the surface, regardless of how it got there and regardless of its physical, chemical, or biological properties.

3. "Site indices" are a local term that have little meaning to an international audience and are not comparable to studies published in other regions. It would be useful if the authors could either clearly define the value of 'site index' in the context of the international literature or select another metric.

Response: Researchers and practitioners in the forestry and soil sciences literature throughout the US and European countries are very familiar with the term site index and it has been used for decades. It is not a local term.

As just two examples,

In Denmark, https://www.tandfonline.com/doi/abs/10.1080/02827580902795036?journalCode=sfor20

In Germany,

https://link.springer.com/article/10.1007/s13595-018-0737-3

And in Sweden,

Tord Johansson. (2011) Site index curves for poplar growing on former farmland in SwedenScandinavian Journal of Forest Research 26:2, pages 161-170. 

I could add several literature references to our paper from other countries that used SI to document this fact, but have decided, for brevity, to add just one (below), which acknowledges that site index is a common term used in forestry literature.

Jarosław Socha, Marcin Pierzchalski, Radomir Bałazy, Mariusz Ciesielski. (2017) Modelling top height growth and site index using repeated laser scanning data. Forest Ecology and Management 406, pages 307-317. 

The paragraph has been changed to:

“Site index (SI) is defined as the total height of a specific species of tree at 50 years age. It is a standard measure used throughout the world as an estimate of the relative productivity of forest land, which is calculated from height measurements of free-growing, uninjured, dominant and co-dominant trees in a fully stocked stand (Carmean et al. 1989; Socha et al. 2017). Site index curves are specific to species, geographic region, and the soil and/or topography (Carmean et al. 1975)…”

4. Much of the background and literature cited focuses exclusively on the Appalachian region, and does not contextualise the results in a global or even national sense. The terminology used throughout is local (understanding it is used within SMCRA and FRA) and not up to step with internationally-accepted norms, see for example the International Standards for the Practice of Ecological Restoration (SER 2016) which clearly articulate the difference between different post-mining activities, and recent commentaries such as "Cross AT, Young R, Nevill P, McDonald T, Prach K, Aronson J, Wardell-Johnson G, Dixon KD. 2018. Appropriate aspirations for effective post-mining restoration and rehabilitation: a response to Kaźmierczak et al. Environmental Earth Sciences 77, 256" which illustrate that "reclamation" is a term no longer widely used by the international scientific community. This study examines rehabilitation, and should utilise the appropriate terminology to reflect this if the study is to be internationally relevant. 

Response: The reviewer is correct is some instances in this comment. The issue of mine soil types for reclamation is a critical problem in the Appalachian region particularly with respect to forestry, but the mine soil type consequences extend to other post-mining land uses, as documented in Skousen and Zipper (2014).

I found the rebuttal referenced above (Cross et al. 2018) to be interesting and an attempt to clarify the terminology of restoration, reclamation, rehabilitation, and revitalization. Again, this has been hashed out in the reclamation literature for more than 50 years with one of the first efforts to define these terms in the book “Reclamation of Drastically Disturbed Lands, 1976, Edited by Frank Schaller and Paul Sutton.

In Chapter 1 of this widely-known and respected book, Thadis Box gives definitions to these terms.

Restoration means “the exact conditions of the site before disturbance will be replicated after disturbance,” and he states, “thus, complete restoration is seldom, if ever, possible.” Restoration is more like “Preservation.”

Further, “Reclamation implies that the site will be habitable to organisms originally present in approximately the same composition and density after the reclamation process has been completed.” This is the terminology essentially used in the Surface Mining Control and Reclamation Act, which must be followed in the US during reclamation of disturbed lands, and which many countries have mirrored in their reclamation standards.

And Box states, “Rehabilitation means that the disturbed site will be returned to a form and productivity in conformity with a prior use plan,” which implies that the site can be reclaimed to other species or conditions that may or may not have been present before mining but has societal value.

The definitions in Chapter 1 (Box) are still relevant and remove the confusion of terminology (at least for me and many of my reclamation colleagues). The fact that the SER (2016) has attempted to clarify these terms (and additional ones such as repair, recovery, revitalization, which are just as nebulous) for their society doesn’t negate the definitions used previously by others, nor are these new definitions widely accepted by the international reclamation community. In fact, the test of time has shown no need to redefine these terms since 1976, not at least until some person in the SER felt they should in 2016. And the fact that the Cross et al. reference is a rebuttal to the definition of terms recommended by the Kaźmierczak et al. article shows that there is still controversy and not consensus on the subject.

Further substantiation could be made by the title of an active group known as the American Society of Mining and Reclamation (ASMR). https://www.asmr.us/

I think the reviewer is incorrect, and his statement that “reclamation is no longer used by the international scientific literature” to be false.

Therefore, we have not changed the manuscript with reference to these comments.

5. The statistical approach is not sufficient, or at least not sufficiently described in the methods. How were linear regression analyses undertaken? With what software? Utilising what factors or variables, and under what conditions? Were underpinning assumptions of normality, heteroscedasticity, and linearity tested and confirmed? Which hypotheses were tested, and how were analyses set up in order to achieve this? Why were ANOVA undertaken on regression coefficients, and not on raw data? What kind of ANOVA? Were post-hoc tests utilised, as it appears from later tables? Were data transformed in any way prior to analysis? If so, why and which transformation? It is inapropriate to simply say "P <0.05" to indicate statistical significance - where are the F values and d.f. for ANOVA, and other regression details from analyses? At the very least, parochial issues aside, the statistical deficiencies must be addressed prior to publication.

Response: We agree that we did not put in sufficient detail in the statistics section. The critical question that we attempted to answer with this study was whether tree growth on mine soils is equal to growth in native forest soils. It is a simple question for which we collected data and analyzed to answer the question. No extensive hypotheses were needed. We analyzed the slopes of lines with the statistical package R, and we have changed the wording in the materials and methods to

“Tree height data from BR and CY mine sites were analyzed using regression with site, year, and species to model annual tree growth rates over time. The slopes of these regression equations on the mine sites for each species were analyzed by ANOVA to determine differences among mine soil types. When significant differences were found with ANOVA (p < 0.05), differences among means were computed with Tukey’s Honestly Significant Difference test at a level of p ≤ 0.05. All statistical analyses were performed using the statistical program R (R Development Core Team, 2012).”         

The figures and tables provide the data to answer this question, which is no, tree growth is lower.

6. Figures are created in excel, and are not of sufficient quality for publication. 

Response:  These were redrawn by our graphics designer at WVU. They are now inserted as tiff files and can be sent as individual files to journal if accepted.

7. Figure and table captions do not contain sufficient information (e.g., details of statistical tests used to infer significance between and among treatments).

Response: The figure and table captions have been redone. Statistical separation information has been placed in footnotes.

8. Many of the references are very old, and in some cases out of date with the current literature. 

Response: Out of the 58 references used in this publication, a quick look at the dates of the references shows:

Publication date from 2010 to 2019: 29 of 58 (50%).

Publication date from 2000 to 2010: 18 of 58 (31%)

Leaving 12 publications cited from before 2000 (12 or 19%).

We feel the references are suitable for this publication and not very old as the reviewer suggests.

9. Both author affiliations are given as 1, despite each having a different affiliation. 

Response: Agree, fixed.

10. I have numerous other comments on specific elements of the manuscript and its content (e.g., grammar and spelling issues throughout), but these are relatively minor in the context of the more significant issues relating to statistical treatment and interpretation of the data. While I am not against the publication of parochial studies, and this study could be published with work, it is not good enough in its current form. Of particular note for me was the authors own comment on line 272: "A better experimental design would have benefited this study".

Response: As for the comment that the reviewer found “grammar and spelling issues throughout,” we have tried to correct any of these errors that we have seen. The other reviewer felt that the English wording and grammar were fine.

We believe this study helps to answer a critical question in the forestry reclamation literature about whether trees growing in reclaimed mine soils are equal to or better than (the wording of SMCRA) trees growing in native soils. While we know of no study performed where trees were planted simultaneously in reclaimed mine soils and undisturbed, cleared areas, we have used SI estimates of growth and the Fernow study as surrogates for growth of trees in native forests during the first 11 years. The reviewer might consider this parochial, but we do not.

Both reviewers felt that the manuscript deserves publication, but that it needed a major revision, which we have done.

Reviewer 2 Report

The topic of the manuscript Early Tree Growth in Brown and Gray Mine Soils on Reclaimed Surface Mines is important and fits into the scope of the Forests journal. The results presented are relevant and interesting, which deserves publication. The assessment procedure and analysis are generally correct. However, the current version of the manuscript requires the following revisions and additions. I would like to point out at two key issues I am concerned about.

I. Although the research is conducted thoroughly and provides valuable data on reclaimed mining sites, it is unclear what was the number of trees used for the study. Lines 127-139 present the description of plots and transects but it is better to give the exact number of studied red oaks, white oaks, and tulip poplars.

II. Please extend the Conclusions with the suggestions based on your findings. What can be done to improve and intensify the reforestation process and which of the factors are most important? This can raise the relevance of your research and outcomes.

III. Please provide a map or a detailed satellite image with the marked study areas to better illustrate the local conditions.

Kindly consider my short remarks that will hopefully improve the manuscript.

1. Line 73: do you mean mine areas within the U.S.?

2. Lines 171: do you mean Table 3?

3. Lines 180-181: ‘Measurements … was not done’?

I sincerely hope you will find my suggestions helpful.

Kind regards,

Reviewer

Author Response

Reviewer 2

1. The topic of the manuscript Early Tree Growth in Brown and Gray Mine Soils on Reclaimed Surface Mines is important and fits into the scope of the Forests journal. The results presented are relevant and interesting, which deserves publication. The assessment procedure and analysis are generally correct. However, the current version of the manuscript requires the following revisions and additions. I would like to point out at two key issues I am concerned about

I. Although the research is conducted thoroughly and provides valuable data on reclaimed mining sites, it is unclear what was the number of trees used for the study. Lines 127-139 present the description of plots and transects but it is better to give the exact number of studied red oaks, white oaks, and tulip poplars.

Response: We thank the reviewer for his kind comments about the study being relevant and interesting and deserves publication.

We agree that an understanding of the number of trees that went into the analyses is important. Therefore, we have inserted an additional sentence to the materials and methods:

“For this study, data for only red oak, white oak, and tulip poplar are discussed. The number of trees sampled for each species to determine average height varied from 37 to 74 for red oak, 37 to 69 for white oak, and 24 to 27 for tulip poplar at the CY and BR sites, respectively (Wilson-Kokes et al. 2013a, 2013b).”

2. Please extend the Conclusions with the suggestions based on your findings. What can be done to improve and intensify the reforestation process and which of the factors are most important? This can raise the relevance of your research and outcomes

Response: We appreciate the reviewer’s comment that the results should be extended. I’ve been doing reclamation work for more than 35 years and I believe the factors that need to be done during reclamation is for us to do the best job we can in landscape design, hydrologic function, soil placement, and revegetation with native species (with species that are aggressive enough to not allow invasives to come in) and then wait and watch. Most of us expect too much too quickly on these reclaimed sites. Reclamation success will be best judged 30 to 50 years after the process began to see the plant community that has developed, the health of the soil and water, and sustainability. Five or 10 years doesn’t generally provide the best judgement for reclamation success.

I have changed the conclusion section to read:

This study was conducted to answer the question “Do native trees planted in mine soils grow at rates equal to or better than trees planted in native forest soils.” Based on the results of this study, the answer is no. Tree heights for three tree species in mine soils after 9 and 11 years were lower than heights estimated from pre-mining native forest soils. Growth rates for red and white oak on brown mine soils had growth rates that were about 50% less than that of trees growing on native soils. Tulip poplar growth rates on brown mine soils were about 70% less than the rates of growth on native soils. In gray mine soils, trees were not growing. Mulching improved the growth rates of trees in mine soils but only slightly better than those growing in brown mine soils. With time, new mine soils will evolve as they weather to finer soil-sized particles, acquire and sequester organic matter, develop greater microbial numbers and diversity, and cycle nutrients. Through this process, mine soils may develop properties over time that are similar to native soils and, with the increased rooting depth and nutrients, may provide conditions where increased tree growth rates above that predicted by SI may be attained. Further studies of tree height 20 to 30 years after planting on these mine soils will help determine the trajectory of growth and will help evaluate improvements in mine soil health over time. The purpose of forestry reclamation is to restore ecosystem diversity, function, and sustainability. In the short term, these purposes were not met, and this study showed that one measure of ecosystem function, tree growth, was 50% lower on reclaimed mine soils than that of native forest soils. However, the placement of suitable soil materials at the surface and the planting of native trees may have begun the process of forest community development, and these newly-formed forest communities may evolve and progress into a functioning and sustainable forest with the passage of time.  

3. Please provide a map or a detailed satellite image with the marked study areas to better illustrate the local conditions.

Response: I have provided google maps of the two sites from which we measured tree heights and soil samples. I don’t think they are very helpful. If readers are interested, maps are available in the two Wilson-Kokes articles. I think they should be deleted but will await the decision of the editor.

4. Kindly consider my short remarks that will hopefully improve the manuscript.

1. Line 73: do you mean mine areas within the U.S.?

Response: Yes, this has been changed.

2. Lines 171: do you mean Table 3?

Response: Yes, this has been changed.

3. Lines 180-181: ‘Measurements … was not done’?

Response: Yes, this has been changed.

Round 2

Reviewer 1 Report

The authors have addressed my major concerns in revising the manuscript, and provide reasonable justification for retaining sections or terminology they prefer to employ. The authors should present all mean values throughout the text instead as mean ± 1 standard error, and include error bars in all figures, but I am happy to recommend this manuscript for publication in good faith that this will occur rather than forcing another round of revisions.

The science and practice of ecological recovery has evolved considerably in the five decades since the work of Schaller and Sutton. The SER International Standards have been developed in consultation with numerous stakeholders at all levels globally (including the US and Canada), and have been written collaboratively and iteratively by many well-respected researchers and practitioners  from these and other countries. While I appreciate the authors points regarding the useage and continuing dialogue surrounding terminology, it is regressive to continue entrenching outdated 50 year old concepts. The science has progressed, and I urge the authors to reconsider their stance for future publications.